# Distribution Pattern of Mangrove Fish Communities in China

**DOI:** 10.3390/biology11121696

**Published:** 2022-11-24

**Authors:** Jinfa Zhao, Chunhou Li, Teng Wang, Chunran Li, Jianzhong Shen, Yong Liu, Peng Wu

**Affiliations:** 1Key Laboratory of South China Sea Fishery Resources Exploitation & Utilization, Ministry of Agriculture and Rural Affairs, South China Sea Fisheries Research Institute, Chinese Academy of Fishery Sciences, Guangzhou 510300, China; 2Guangdong Provincial Key Laboratory of Fishery Ecology Environment, Guangzhou 510300, China; 3Observation and Research Station of Pearl River Estuary Ecosystem, Guangdong Province, Guangzhou 510300, China; 4College of Fisheries, Huazhong Agricultural University, Wuhan 430070, China; 5Key Laboratory of Efficient Utilization and Processing of Marine Fishery Resources of Hainan Province, Sanya Tropical Fisheries Research Institute, Sanya 572018, China

**Keywords:** mangrove fishes, reef fishes, fish community, feeding habit, habitat type, endangered degree

## Abstract

**Simple Summary:**

The mangrove ecosystem is an important resource for the survival of thousands of species of animals and plants. It has high ecological and economic value and has made great contributions to the protection and maintenance of marine and terrestrial ecological environment. However, due to climatic and human factors, the mangrove ecosystem has been destroyed and its ecological functions have been damaged. As an important indicator species, fishes have also been affected, with changes in their population and nutritional structure. This study summarized the mangrove fish in China and compiled a relatively complete list of mangrove fish. In addition, the biogeographic characteristics of Chinese mangrove fish were analyzed, filling the gap of national mangrove fish distribution data. Completion of this study contributed to a better understanding of the role of mangroves in maintaining the stability and diversity of fish communities, which was of great value for the management and conservation of regional and global biodiversity.

**Abstract:**

Mangroves are among the most productive marine and coastal ecosystems and play an important role in maintaining the stability and diversity of fish communities. To explore the structure of mangrove fish communities in China, we compiled previous studies, monographs, and two databases on 54 mangrove areas published in the past 30 years. Mangrove fish communities in China comprised Osteichthys (597 species) and Chondrichthyes (14 species), representing 611 species in 344 genera, 117 families, and 28 orders. Perciformes were the predominant taxon, with 350 species in 52 families, accounting for 57% of the total species richness. Reef fish accounted for 29.62%. With regard to feeding groups, there were 328 carnivorous species (53.68%), 214 omnivorous species (35.02%), 41 herbivorous species (6.71%), and 28 detritivores species (4.58%). Classified by body size, 57.61% were small-sized, 24.22% medium-sized, and 18.17% were large-sized fishes. A total of 5.23% (32 species) of these mangrove fish are currently on IUCN red lists, i.e., 2 species are critically endangered, 4 are endangered, 12 are vulnerable, and 14 are near threatened. Cluster analyses shows that Chinese mangroves fish were divided into two categories, i.e., coastal mangrove and island mangrove type. This is closely related to the distribution of reef fish. Moreover, the number of fish species showed a strong positive correlation with mangrove area, but not with latitude. The main reasons may be the subtropical and tropical geographic locations, as well as the characteristics of the South China Sea and the Taiwan Warm Current. The size and integrity of mangrove area are crucial to the local ecosystems; thus, protecting and restoring mangroves is of great significance to large-scale ecosystem-stability and local biodiversity.

## 1. Introduction

Mangroves play an important role in national and regional economic development, supporting 30% of global commercial fishery [1,2]. Mangrove ecosystems are characterized by high productivity, high biodiversity, and complex structures. These habitats provide shelter, feeding and spawning grounds for fish, and doubling the biomass of some commercially important fish species [3,4,5]. In addition, mangroves also make considerable contributions to ecosystem stability and functioning, which is crucial for the safety of human life in such areas. As a barrier protecting coastal systems, mangroves play an important role in reducing wave impact, filtering harmful substances, and sequestering carbon [6,7,8,9]. Globally, mangroves are mainly distributed in four tropical geographic regions, i.e., the Indo-West Pacific, Eastern Pacific, Western Atlantic, and Eastern Atlantic [10]. Among them, the Asian region of the Indo-West Pacific comprises the largest and most diverse mangrove areas, accounting for about 42% of the global mangrove area [11].

Due to human impact, especially wetland reclamation, pollution through aquaculture, and urbanization, mangroves have experienced severe degradation and area decline. Global mangrove areas are estimated to have decreased by 30–50% in the past 50 years [12,13]. The Sustainable Development Goals established by the United Nations are closely related to the protection of mangroves, and China has also launched a series of projects for the protection and restoration of mangroves [12,14]. It is worth noting that not only trees but also important ecological functions such as fish, waterbirds, benthic animals, and mangrove-associates should be considered for the restoration of mangroves [15,16,17].

Fish are an important component of mangrove ecosystems where they occur at multiple trophic levels. The fish community structure can reflect major and minor changes in the ecological environment, and it is an important indicator of mangrove ecosystem stability and functioning [18,19,20]. However, the current status of mangrove fish communities does not make for optimistic prospects. Overexploitation and disproportionate targeting of fish at high trophic levels will affect fish population structures and life history process [21]. The trophic imbalance of mangrove fish communities and combined adverse effects will continue to affect ecosystem health and reduce ecosystem stability [22]. Approximately 4.1 million people work in mangrove-related fishery, of which Asia has the highest fishing intensity [23]. Although research on Chinese mangrove fish has been initiated, it is typically limited to small areas, for example, Hainan Wenchang [24], Quanzhou Bay [25], Beilun Estuary [26], Maowei Sea [27]; however, large-scale processes, structures, and changes in Chinese mangrove waters are unclear. Therefore, the research on fish in mangrove areas in China has important theoretical and practical significance for the protection of mangrove ecosystems in Asia and in the world.

This study was conducted to describe the structural characteristics and distribution pattern of Chinese mangrove fish communities. Specifically, (1) we summarized the species composition, dietary characteristics, habitat type, and endangerment degree of Chinese mangrove fish; (2) we studied the fish community structure in 54 mangrove areas in China on a large spatial scale and analyzed the distribution pattern of mangrove fish in various geographic regions of China; (3) we assessed the extent to which mangrove area and latitude predict fish communities in Chinese mangroves, and we examined the reasons for changes in fish community structures over a gradient. These insights contribute to a better understanding of the role of mangroves in maintaining the stability and diversity of fish communities, with important outcomes regarding the management and conservation of regional and global biodiversity [22,28].

## 2. Materials and Methods

### 2.1. Study Areas

The study area comprised documented Chinese mangrove areas. The currently known distribution locations of Chinese mangrove fish ranged from the southernmost Qingmei Port to the northernmost Tanshui River, with a total of 54 mangrove areas (Table 1, Figure 1). The provinces or administrative regions involved included Guangdong, Guangxi, Hainan, Fujian, Hong Kong, Taiwan, and Macau. To achieve general representativeness of the data and analysis accuracy, we divided the 54 mangrove areas into 16 regions by geographic location, and the areas with less survey data were excluded. These areas experience a subtropical and tropical marine monsoon climate with an annual average temperature of 21–25 °C [29]. The tides tend to vary, with regular semi-diurnal tides in Fujian, irregular semi-diurnal tides in Guangdong, regular diurnal tides in Guangxi, and irregular semi-diurnal tides and irregular diurnal tides in Hainan and Taiwan. Water salinity shows considerable variation owing to changes in rainfall and river runoff, and it is higher in summer and lower in winter, ranging from 0.7 to 34.4‰ [24,30].

### 2.2. Data acquisition

#### 2.2.1. List of Mangrove Fish Species

First, all the mangrove areas in China were identified. Second, all the research data on mangrove fish in China were collected. Third, classified according to the geographical position, the size and latitude of the mangrove areas were determined and the fish list of different regions compiled. Fourth, the data of habitat type, feeding habits, conservation status, and body size of various fish were found according to the database. Finally, the list and related data of mangrove fish in China were obtained. Statistical data on species composition, morphological characteristics, and community structure of fishes in mangrove areas of China were used from historical research. Fish data were obtained from two sources: (1) published studies, regional inventories, reports, and monographs (Table 1); (2) the Fishbase Database (https://www.fishbase.se/search.php, accessed on 26 October 2022) and the Fish Database of Taiwan (https://fishdb.sinica.edu.tw/chi/home.php, accessed on 26 October 2022). Synonyms in the compiled fish list were adjusted, and undetermined species were eliminated to ensure accuracy of data analysis, thus a comparably complete list of Chinese mangrove fish was produced (Appendix A). 

#### 2.2.2. Habitat Types

The habitat types of fish species on the final list were obtained from the Fishbase Database (https://www.fishbase.se/search.php), and reef-associated fish were analyzed separately.

#### 2.2.3. Feeding Habits

Fish were assigned to four feeding groups, namely herbivores, carnivores, detritivores, and omnivores [83]. 

#### 2.2.4. Conservation Status

The conservation status of each fish species was obtained from the International Union of Conservation of Nature and Natural Resources Red List (IUCN Red List) (https://www.iucnredlist.org/, accessed on 26 October 2022), using the categories critically endangered (CR), endangered (EN), vulnerable (VU), near threatened (NT), least concern (LC), data deficient (DD), and not evaluated (NE).

#### 2.2.5. Body Size

The maximum length of fish (regardless of sex) is obtained from the Fishbase Database (https://www.fishbase.se/search.php). According to the maximum total length, fish were considered small-sized (maximum length < 35 cm), medium-sized (65 cm > maximum length ≥ 35 cm), or large-sized (maximum length ≥ 65 cm) [84].

#### 2.2.6. The size and Latitude of the Mangrove

The size and latitude of the mangrove study areas were derived from published studies, regional inventories, reports, and monographs. In addition, the mangrove size data was normalized in logarithmic form during the analysis [85].

### 2.3. Data Analyses

PRIMER 5.0 and R v.4.2.1 (package ggpolt2) [86] (R: A language and environment for statistical computing. R Foundation for Statistical Computing, Vienna, Austria. URL https://www.R-project.org/, accessed on 26 October 2022.) were used for cluster analysis (CLUSTER), nonparametric multidimensional analysis (NMDS), and species contributions to similarity (SIMPER) analysis. SPSS v.24 (IBM, Armonk, NY, USA) was used for Pearson correlation analysis, Spearman correlation analysis and similarity analysis (ANOSIM). The fish community structure was investigated using methods such as graphical analysis and multivariate analysis with PRIMER 5.0 and R 4.2.1 software. The number of fish species per mangrove area was counted, and Bray–Curtis similarity matrices were produced based on the number of fish species per order and through square root-transformation. Then, clustering using CLUSTER and nonparametric multidimensional scaling (NMDS) were used for further analyses. After several iterations, the optimal result was obtained. The judgment standard was lower stress value. It is generally believed that stress < 0.2 has explanatory significance [87]. The difference of fish community composition in different mangrove regions was analyzed as described previously [63]. SPSS v. 24 was used to analyze the correlation between the area and latitude of each mangrove area and the community characteristics. Before statistical analyses, the data were tested for homogeneity of variance and normal distribution. When these conditions were met, Pearson’s correlation analysis was performed, otherwise Spearman’s correlation analysis was used. Linear regression models were fitted to analyze the relationship between mangrove area and latitude and fish community characteristics (including: number of species, habitat types, feeding habits and body size) [88]. Analysis of similarities (ANOSIM) was used to test the significance of differences in fish composition in different regions, and species contributions to similarity (SIMPER) analysis was used to calculate the similarity of fish composition in different regions and the average contribution rate of species to similarity [89].

## 3. Results

### 3.1. Fish Community Composition

Excluding undetermined species, Chinese mangrove fish were found to belong to the classes Osteichthys (597 species) and Chondrichthyes (14 species), with a total of 611 species in 344 genera, 117 families, and 28 orders. Perciformes were the predominant taxon, with 350 species in 52 families, accounting for 57% of the total species richness. The second most numerous orders were Clupeiformes (43 species), Pleuronectiformes (34 species), Anguilliformes (26 species), Tetraodontiformes (24 species), Mugiliformes (19 species), Scorpaeniformes (20 species), Beloniformes (17 species), and Cypriniformes (15 species). The remaining orders comprised fewer than 10 species, each, i.e., Torpediniformes, Myliobatiformes, Orectolobiformes, Rajiformes, Carcharhiniformes, Cichliformes, Gasterosteiformes, Myctophiformes, Pegasiformes, Elopiformes, Synbranchiformes, Gobiesociformes, Cyprinodontiformes, Stomiiformes, Siluriformes, Gonorhynchiformes, Aulopiformes, Gadiformes, and Atheriniformes (Figure 2).

The number of fish species in each mangrove regions were the largest in Western Guangdong, with 263 species. Followed by Zhangzhou, Fujian, Haikou, Hainan, and Wenchang, Hainan, more than 150 species. However, there were fewer fish species in Qinzhou, Guangxi, Fangchenggang, Guangxi, Dongxing, Guangxi, Pearl River Delta, Hong Kong, Sanya, Hainan, and Eastern Guangdong, all fewer than 100 species (Table 1, Figure 1). The composition of fish communities differed between mangrove regions. The first dominant order of different mangrove fish communities was the same in each study area, i.e., Perciformes, accounting for more than 53.2% (Figure 2). However, the second dominant orders differed, i.e., Clupeiformes in Wenchang, Hainan, Haikou, Hainan, Qinzhou, Guangxi, Beihai, Guangxi, Pearl River Delta, Zhangzhou, Fujian, Tainan, Taiwan, and Taipei, Taiwan, Tetraodontiformes in Sanya, Hainan, and Mugiliformes in Fangchenggang, Guangxi, Dongxing, Guangxi, Hong Kong, and Quanzhou, Fujian (Figure 2). The composition of fish orders varies in different mangrove regions. Moreover, Myctophiformes, Torpediniformes, Pegasiformes, Synbranchiformes, Gobiesociformes, Stomiiformes, Gonorhynchiformes, Orectolobiformes, Gadiformes, and Rajiformes are the least distributed orders and only exist in one region (Figure 2).

### 3.2. Fish Community Characteristics in the Study Regions

#### 3.2.1. Habitat Type

According to the Fishbase database, the reef fish accounts for 29.62% of the mangroves in China. Among the different mangrove regions, the proportion of reef fish in Hong Kong was the highest (45.28% of all fish species in the region), followed by Wenchang, Hainan (39.88%), Sanya, Hainan (36.73%), Tainan, Taiwan (35.11%), Eastern Guangdong (33.33%), and Taipei, Taiwan (30%); in the remaining regions, reef fish accounted for less than 30% (Figure 3A).

#### 3.2.2. Feeding Habits

Throughout the range of mangroves in China, carnivorous fish accounted for 53.68% of all species, followed by omnivores (35.02%), herbivores (6.71%), and detritivores (4.58%). The proportion of herbivores and detritivores fishes was similar and significantly smaller than that of carnivorous and omnivores fishes. Thus, dietary characteristics of fish communities were similar in all mangrove regions (Figure 3B).

#### 3.2.3. Conservation Status of Fishes in Different Regions

After identifying the conservation status of fish species in the Fishbase Database, 5.23% (32 species) of the species living in Chinese mangroves were threatened, of which critically endangered species accounted for 0.33% (2 species), endangered for 0.65% (4 species), vulnerable for 1.96% (12 species), and near threatened species accounted for 2.29% (14 species). Similar patterns were detected in various mangrove regions, where the proportion of endangered fish is high: Zhangzhou, Fujian (7.58%), Dongxing, Guangxi (7.04%), Fangchenggang, Guangxi (6.85%), Western Guangdong (6.46%), Zhanjiang, Guangdong (5.44%), and Tainan, Taiwan (5.34%). The proportion of endangered fish in other regions was slightly lower, yet considerable, such as: Haikou, Hainan (4.09%), Sanya, Hainan (4.08%), Hong Kong (3.77%), Beihai, Guangxi (3.54%), Wenchang, Hainan (3.07%), Taipei, Taiwan (3.00%), Quanzhou, Fujian (2.75%), Qinzhou, Guangxi (2.19%), and Pearl River Delta (1.45%) (Figure 3C).

#### 3.2.4. Fish Body Size

Overall, the proportion of small-sized fishes was the highest (57.61%), followed by that of medium-sized fishes (24.22%) and large-sized fishes (18.17%). In each mangrove area, the fish community was dominated by small-sized fishes (>45%), and the proportion of large-sized fishes was the smallest (approximately 20%) (Figure 3D).

### 3.3. Distribution Patterns of Fish Communities

Cluster analysis of fish communities in Chinese mangroves showed that at a similarity level of 30%, 16 mangrove regions were divided into 4 groups; Eastern Guangdong only was assigned to group A; Sanya, Hainan, and Hong Kong were assigned to group B; Wenchang, Hainan, Taipei, Taiwan, and Tainan, Taiwan were assigned to group C; Quanzhou, Fujian, Haikou, Hainan, Western Guangdong, Zhanjiang, Guangdong, Zhangzhou, Fujian, Pearl River Delta, Fangchenggang, Guangxi, Dongxing, Guangxi, Qinzhou, Guangxi, and Beihai, Guangxi were assigned to group D (Figure 4). The results of the NMDS analysis (Figure 5) also supported the clustering (stress = 0.117 < 0.2). The SIMPER similarity percentage reflects the intra-group similarity and inter-group dissimilarity, as well as key species with larger contribution rates. As group A comprised only one region, SIMPER intragroup similarity analysis was performed only on groups B, C, and D. The results of intra-group similarity showed that the intra-group similarity of group B was 31.37%, and the highest contribution rate of *Periophthalmus modestus* was 93.75%; the similarity within group C was 34.54%, of which *Acentrogobius caninus* had the highest contribution rate of 90.15%; the similarity within group D was 40.30%, among which *Megalops cyprinoides* had the highest contribution rate of 90.13%. The results of the SIMPER between groups show that, except for the lower degree of dissimilarity between groups C and D (72.71%), dissimilarity between groups generally exceeded 75% (Table 2). The ANOSIM test results were comparable, with a Global R range of 0–1 (the closer R is to 1, the greater the difference). The R value between groups C and D was 0.71, whereas that between the other groups were larger, indicating significant differences between the four groups (Table 2).

### 3.4. Correlation of fish Community Characteristics with Size and Latitude

The total number of fish species was positively correlated with the size of mangrove size (*r* = 0.71, *p* < 0.01) (Figure 6A), while the proportion of reef fish showed a significant negative correlation with mangrove size (*r* = −0.46, *p* < 0.01) (Figure 6B). Further detailed analyses of the fish community results showed that the size of mangroves was also correlated with fish feeding habits and was significantly positively correlated with the proportion of carnivorous fish (*r* = 0.42, *p* < 0.05) (Figure 6C). The proportion of omnivores was negatively correlated (*r* = −0.44, *p* < 0.01) (Figure 6D). The correlations of mangrove size with the proportion of herbivores and that of detritivores were not significant (*r* = 0.15 and *r* = −0.16, respectively) (Figure 6E,F). With the increase of mangrove size, the proportion of fish size changed, and the proportion of small-sized fish and large-sized fish showed a decreasing, yet non-significant, trend (*r* = −0.10 and *r* = −0.24, respectively) (Figure 6G,I), while there was an increasing, yet non-significant, trend in medium-sized fish (*r* = 0.22) (Figure 6H).

The correlation of fish community indicators with latitude of mangroves and was less pronounced than that with mangrove size. With respect to fish feeding habits, the proportion of omnivores was positively correlated with latitude (*r* = 0.38, *p* < 0.05) (Figure 7D), while the proportions of carnivores, detritivores, and herbivores showed non-significant trends of correlation with mangrove latitude (*r* = −0.22; *r* = −0.12; and *r* = 0.15, respectively) (Figure 7C,E,F). Other fish community indicators (number of species, habitat types, and fish size) were also not correlated with mangrove latitude (Figure 7A,B,G–I).

## 4. Discussion

### 4.1. Mangrove Fish Community Composition in China

With this study, we summarized the population structure of mangrove fish in China. The number of mangrove fish species in China is markedly higher than that in other countries, including New Caledonia (262 species) [90], the Sultanate of Oman (177 species) [91], and Vietnam (258 species) [92]. This high mangrove fish diversity is mostly attributable to the expansive coastline and advantageous geographical location [93]. In the present study, Perciformes was the predominant order in all regions, which was consistent with the results of studies conducted in other mangrove systems, such as in the Sultanate of Oman [91], the Philippines [94], and Panama [95]. In addition, we found that the number of fish species in the order Clupeiformes by far exceeded that in other orders, and most of them were tenacious, euryhaline, and eurythermal [95,96]. The numbers of herbivorous and detritivores fish species were comparably lower in the mangrove regions of China, which is consistent with low abundances but high biomass of herbivorous fish in mangroves in other geographical regions [84,97,98]. Herbivorous fish play a connecting role in their respective ecosystems, as they control the growth of large algae through grazing (i.e., top-down effects) and they regulate the populations of larger carnivorous fish as their prey (bottom-up effects). However, low abundances of herbivorous fish species may result in low elasticity of ecosystems [99], as herbivore populations can buffer adverse effects that disturb ecosystem functioning at lower and higher trophic levels. Moreover, numerous marine fish species and their respective habitats are severely threatened by climate change, overfishing, alien invasion, and environmental pollution [100]. Since 1600, 90 non-indigenous species have been introduced into the South China Sea, 32 of which are fish, some of which have successfully invaded brackish ecosystems, such as mangroves and estuarine wetlands. The primary pathways of introduction are through aquaculture, followed by shipping, ecological restoration, and biocontrol [101,102,103]. Mangroves constitute important habitats of many endangered fish. A total of 32 threatened fish species in Chinese mangroves, accounting for 5.23% of the total fish species diversity. However, many mangroves have been severely degraded in recent years due to overexploitation and pollution (including heavy metals, PCBs, microplastics, etc.) [12,104]. Therefore, awareness must be raised for the protection and restoration of mangrove habitats.

### 4.2. Fish Community Distribution Patterns among Different Mangrove Regions

The 16 mangrove regions examined in the current study were divided into four groups through cluster analysis at fish species level. Mangrove fish communities in China were thus divided into two categories: coastal mangrove type (groups A and D) and island mangrove type (groups B and C) [105].The proportion of reef fish made a considerable contribution to the results, and they represented a larger proportion in fish communities of island mangroves (35%) than in those of coastal mangroves (20%). This is likely due to the stronger ocean circulation, clearer water, and more suitable environment for coral growth in sea areas surrounding island reef mangroves, which promotes connectivity between mangrove and coral reef ecosystems [106,107]. Therefore, coastal mangroves and reef mangroves are characterized by their respective unique fish fauna. It is worth noting that according to the investigation on habitat types of mangrove fish in China, reef fishes accounted for 29% of the total. In particular, Hong Kong (45%) had the largest proportion of reef fishes in mangrove regions. Meanwhile, the fish community was generally dominated by small-sized fishes, accounting for 57% of the total number. This result was consistent with those of previous studies on mangroves as fish shelters, feeding grounds, and spawning grounds, and it further confirmed the importance of connectivity of mangroves with coral reefs [108,109,110]. Connectivity of mangrove and coral reef ecosystems has been demonstrated in studies in Europe [111], East Asia [54], Southeast Asia [112], and the Southwest Atlantic [113]. Studies in Caribbean waters found that the biomass of several species in coral reefs linked to mangroves has more than doubled [114]. Du, Xie, Wang, Chen, Liu, Liao, and Chen [54] described the connectivity of fish communities on the mangrove-seagrass-coral reef continuum in Wenchang, China, and emphasized the importance of mangroves as habitats for juvenile fish. Through such connectivity and exchange functions, mangroves can produce and export large quantities of fish, including commercially important species; therefore, stable functioning of mangrove ecosystems facilitates stability of fishery resources and ensures diversity of fish communities in mangrove and coral reef ecosystems [115]. 

The distribution patterns of fish are the result of long-term adaptation of fish populations to the environment, and the composition of fish communities across different areas with similar habitat types is remarkably similar [116]. In coral reefs, such similarity of fish communities between geographically distant reef areas is higher than that between geographically close reef areas and shelf areas [117]. Comparably, even though the mangrove regions of Wenchang City and Taiwan are geographically distant from each other, both are typical mangrove ecosystems with strong connectivity with coral reefs, and both are affected by the South China Sea Warm Current and the Kuroshio branch [54,118]. The environmental conditions of these two mangrove sites are relatively similar, which may explain why their fish communities are also relatively similar, and they were categorized in the same group. In addition, the Haikou mangrove is a reef mangrove, which is geographically close to Zhanjiang, and these two sites are connected through hydrology and human transport activities. Fish inhabiting these two sites exhibit habitat migration and elicit close genetic exchange at some stage in their life cycle, which may explain why the Haikou mangroves are divided into separate inshore mangroves [119].

### 4.3. Fish Community Characteristics in Different Size and Latitudes

We found that the number of fish species was positively correlated with mangrove size, which is consistent with respective findings in Malaysia [120], Indonesia [120], Vietnam [2], and the Caribbean [115]. Habitat heterogeneity is an important factor for biodiversity [121], and with increasing habitat complexity, the ecological niche breadth increases and thus the number of species [122,123]. Mangroves are highly complex ecosystems, mostly because (1) the forms of plants in mangrove regions are diverse; (2) fallen leaves, sediments, and tides can substantially and rapidly change the turbidity of the water; and (3) owing to inflow from rivers, the water salinity range may vary considerably [108,110,124]. Vaslet et al. [125] investigated the habitat trophic-level structure of the Guadeloupe mangrove coastline and found that food availability strongly affects mangrove fish community composition. Fallen leaves and sediments can provide large amounts of nutrients and constitute rich food sources for some fish species through the detritus food chain [10]. Wu, et al. [126] collected mangrove species data at 70 sites in China and found that mangrove species composition exhibited a nested structure. Nesting of species communities was significantly correlated with habitat size, and nesting patterns were frequent on islands [127,128]. Therefore, the focus of conservation efforts should be placed on protecting mangroves covering large size, which provide more habitat space and thus sustain more fish [63,129]. Londono et al. [130] investigated a mangrove size in the Southern Caribbean and found that mangrove size was the main factor affecting fish richness, which was significantly and positively correlated with the catch of common fish. However, it is insufficient to rely solely on mangrove size when judging habitats regarding the importance of protection. Moreover, functional integrity and plant density are important factors affecting fish diversity. Tran and Fischer [2] studied the relationship between habitat fragmentation and fish diversity in mangroves of CaMau Province, Vietnam, and fish diversity in mangroves with higher fragmentation was 1.78-fold lower than that in less fragmented mangroves. Sitorus et al. [131] found a very strong correlation between mangrove density and fish diversity in mangroves of North Sumatra. Taken together, protection and restoration of complete large-size mangroves were key to protecting the fish resources in these regions.

Latitude is frequently considered an important variable affecting species richness, and fish richness changes over the latitude [132,133]. The correlation between species composition and latitude has been confirmed in the Great Barrier Reef [134], Mexico [135], and California [136]. However, in our study, the number of species of mangrove fishes was not correlated with latitude, probably because the range of latitude was not sufficiently large for such effects to occur. A survey by Zintzen, et al. [137] showed that fish community structures in New Zealand were strongly correlated with latitude (29.15–50.91 degrees S). Holland et al. [138] studied the relationship between the trophic structure of coral reef fish in eastern Australia and latitude (29–44 degrees S) and found that fish biomass decreased towards higher latitude. However, in the current study, the latitude of the examined systems was 18–25 degrees S, thus covering a comparatively narrow range. Most importantly, water temperature is lower at higher latitudes, which also affected fish communities. Travers, Potter, Clarke, Newman, and Hutchins [124] found that changes in the structure of inshore fish populations on soft substrates and at coral reefs on the tropical west coast of Australia were driven by differences in water temperature across latitudes. However, mangroves in China are located in the tropics and subtropics. Due to the South China Sea Warm Current and the Taiwan Warm Current, the average water temperature in the mangrove regions were 21–25 °C [29,30,124]. Therefore, the species composition of mangrove fish in Chinese waters is less affected by latitude differences. Similar results were produced in previous studies. For example, Kulbicki et al. [139] analyzed the biogeography of Chaetodontidae in the western and central Pacific areas and found that Chaetodontidae diversity was not significantly correlated with latitude. Roberts surveyed coral reef fish communities in the Saudi Arabian Red Sea between 26.8 to 18.6 degrees N and found only subtle changes in fish communities along the latitude gradient, which were closely related to environmental conditions. More challenging environmental conditions may increase homogeneity among species communities [140].

There are two major limitations in this study that could be addressed in future research. First, the study focused on the distribution of mangrove fish in China at large spatial scales. Only two variables, which are latitude and mangrove size, were analyzed and small-scale variables for the potential effect of fish communities including temperature, tides levels, salinities, depths, mangrove community composition, human influences, pollutant discharge, and urbanization were ignored. Therefore, it is of great significance to study these small-scale variables in future research. Second, this study did not analyze the changes of mangrove fish communities between different time periods. Since the data span is 30 years, it is of great significance to analyze the changes of mangrove fish communities in different time periods in future studies.

## 5. Conclusions

We summarized the distribution patterns of mangrove fish in China, and we describe the biogeographic characteristics of Chinese mangrove fish and present a relatively complete list of Chinese mangrove fish. The insights produced here will be of importance for decision makers with respect to regional and global mangrove conservation and restoration. Further, our results fill a gap in national mangrove fish species distribution data. Considering the grave adverse effects of anthropogenic disturbance and climate change on these sensitive ecosystems, our study constitutes an important reference for the protection and restoration of global mangrove ecosystems, and it provides data support for the protection of mangrove fish communities.

## Figures and Tables

**Figure 1 biology-11-01696-f001:**
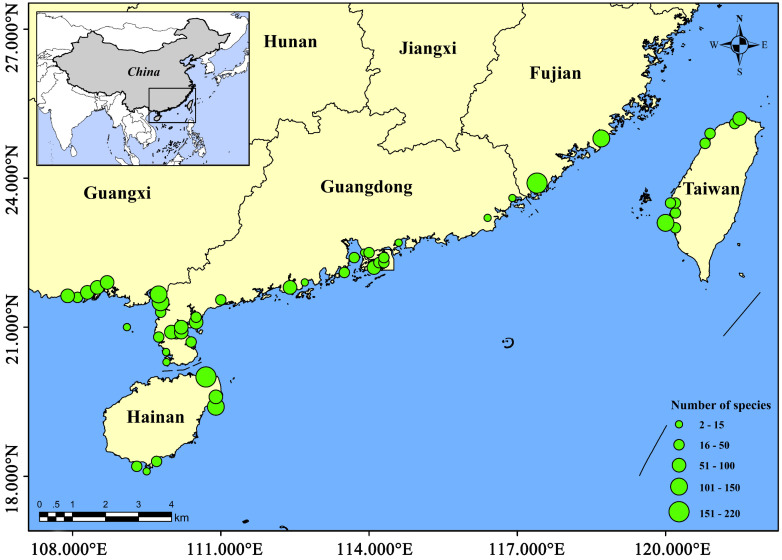
Distribution of fish populations in different study areas. The green circle in the figure is the number of fish species.

**Figure 2 biology-11-01696-f002:**
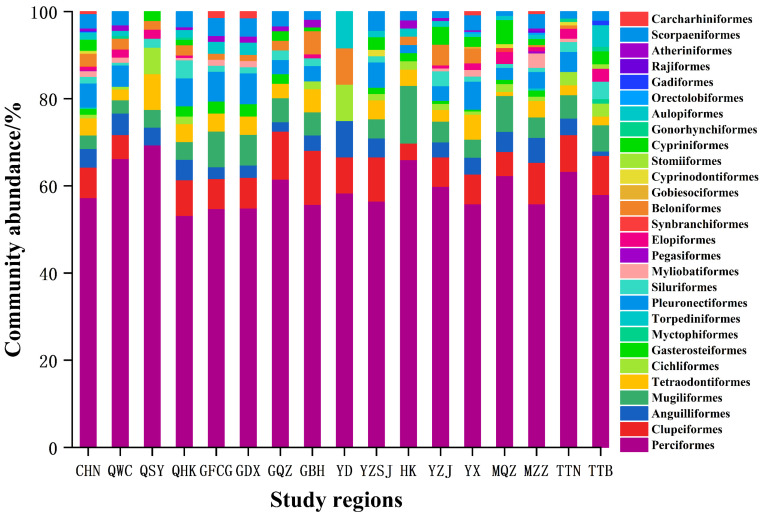
Relative abundance of fish community composition at the order levels across different mangrove regions. CHN: CHINA; QWC: Wenchang, Hainan; QSY: Sanya, Hainan; QHK: Haikou, Hainan; GFCG: Fangchenggang, Guangxi; GDX: Dongxing, Guangxi; GQZ: Qinzhou, Guangxi; GBH: Beihai, Guangxi; YD: Eastern Guangdong; YZSJ: Pearl River Delta; HK: Hong Kong; YX: Western Guangdong; YZJ: Zhanjiang, Guangdong; MQZ: Quanzhou, Fujian; MZZ: Zhangzhou, Fujian; TTN: Tainan, Taiwan; TTB: Taipei, Taiwan.

**Figure 3 biology-11-01696-f003:**
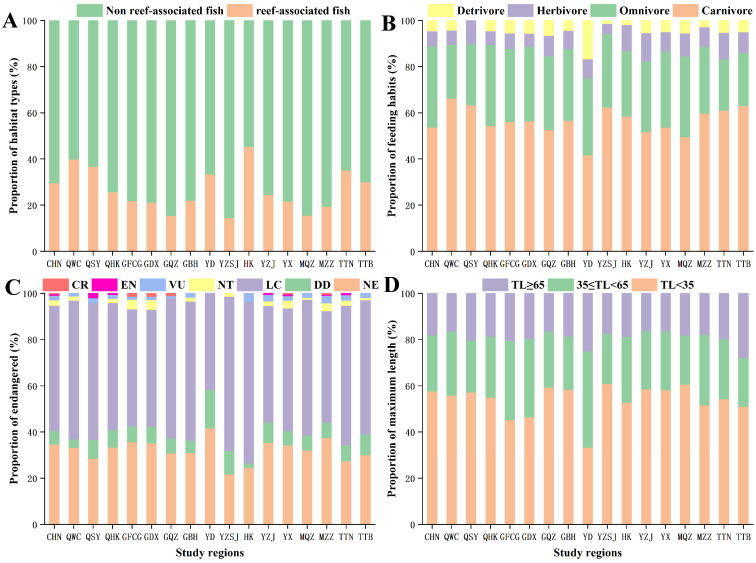
Distribution characteristics of fish community structure in different regions. (**A**) Habitat types of fishes in different regions. (**B**) Feeding habits of fishes in different regions. (**C**) Endangered of fishes in different regions. (**D**) Maximum length of fishes in different regions. CHN: CHINA; QWC: Wenchang, Hainan; QSY: Sanya, Hainan; QHK: Haikou, Hainan; GFCG: Fangchenggang, Guangxi; GDX: Dongxing, Guangxi; GQZ: Qinzhou, Guangxi; GBH: Beihai, Guangxi; YD: Eastern Guangdong; YZSJ: Pearl River Delta; HK: Hong Kong; YX: Western Guangdong; YZJ: Zhanjiang, Guangdong; MQZ: Quanzhou, Fujian; MZZ: Zhangzhou, Fujian; TTN: Tainan, Taiwan; TTB: Taipei, Taiwan.

**Figure 4 biology-11-01696-f004:**
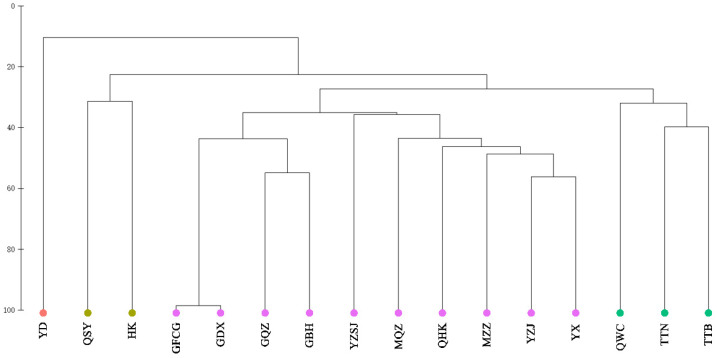
Cluster diagram of fish community structure in mangrove of China. YD: Eastern Guangdong; QSY: Sanya, Hainan; HK: Hong Kong; GFCG: Fangchenggang, Guangxi; GDX: Dongxing, Guangxi; GQZ: Qinzhou, Guangxi; GBH: Beihai, Guangxi; YZSJ: Pearl River Delta; MQZ: Quanzhou, Fujian; QHK: Haikou, Hainan; MZZ: Zhangzhou, Fujian; YZJ: Zhanjiang, Guangdong; YX: Western Guangdong; QWC: Wenchang, Hainan; TTN: Tainan, Taiwan; TTB: Taipei, Taiwan.

**Figure 5 biology-11-01696-f005:**
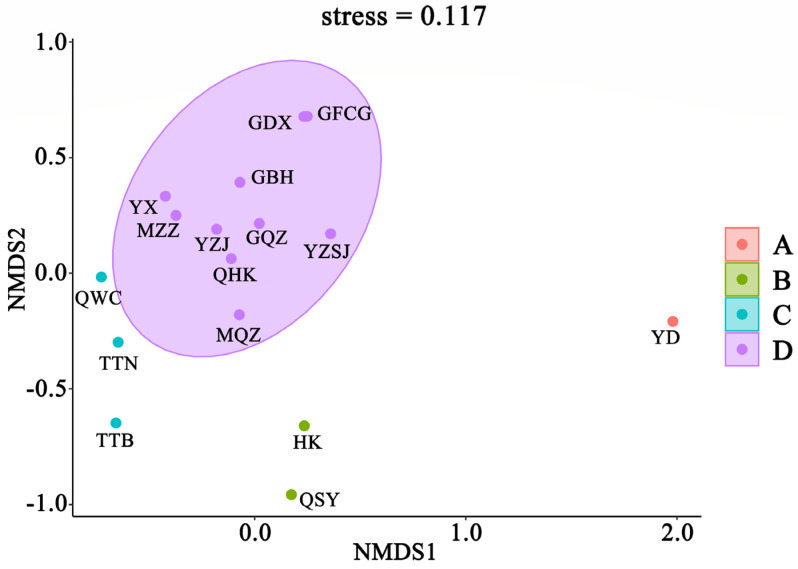
Nonparametric multivariate sequence of fish community structure in mangrove of China. YD: Eastern Guangdong; QSY: Sanya, Hainan; HK: Hong Kong; GFCG: Fangchenggang, Guangxi; GDX: Dongxing, Guangxi; GQZ: Qinzhou, Guangxi; GBH: Beihai, Guangxi; YZSJ: Pearl River Delta; MQZ: Quanzhou, Fujian; QHK: Haikou, Hainan; MZZ: Zhangzhou, Fujian; YZJ: Zhanjiang, Guangdong; YX: Western Guangdong; QWC: Wenchang, Hainan; TTN: Tainan, Taiwan; TTB: Taipei, Taiwan.

**Figure 6 biology-11-01696-f006:**
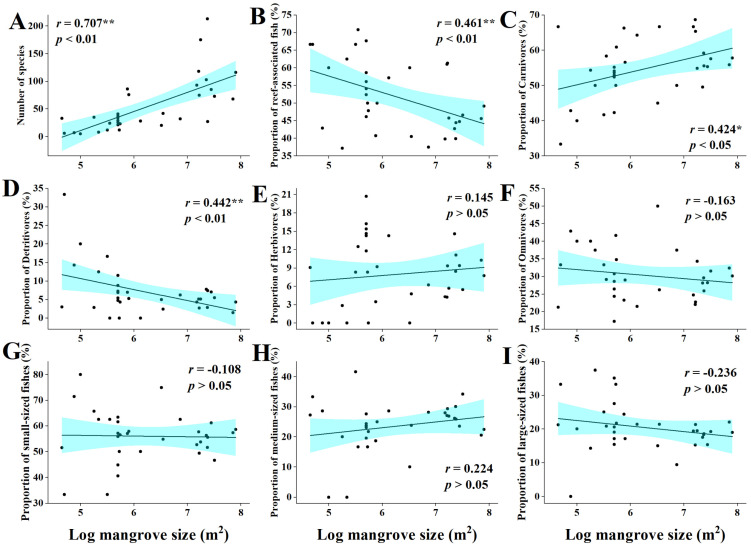
Relationship between mangrove size and characteristics of fish communities. (**A**) The relationship between number of fish species and the mangrove size. (**B**) The relationship between reef-associated fish and the mangrove size. (**C**) The relationship between carnivores and the mangrove size. (**D**) The relationship between detritivores and the mangrove size. (**E**) The relationship between herbivores and the mangrove size. (F) The relationship between omnivores and the mangrove size. (**G**) The relationship between small-sized fishes and the mangrove size. (**H**) The relationship between medium-sized fishes and the mangrove size. (I) The relationship between large-sized fishes and the mangrove size. (**: Extremely Significant, *p* < 0.01. *: Significant, *p* < 0.05).

**Figure 7 biology-11-01696-f007:**
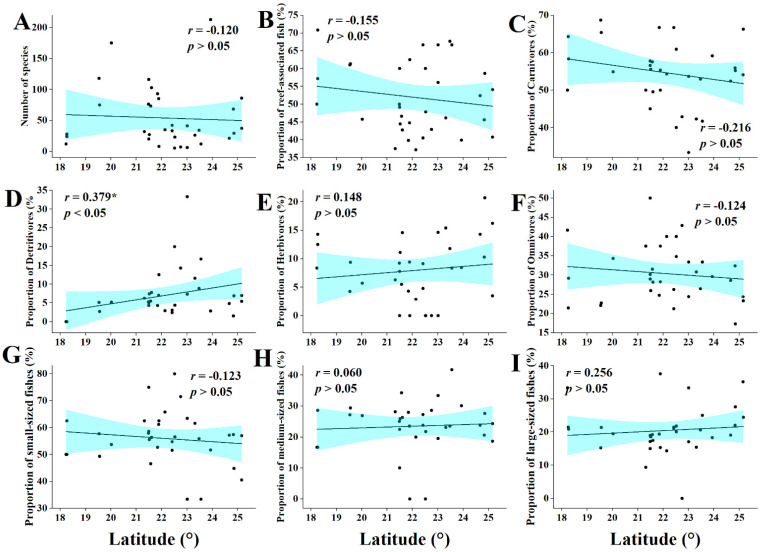
Relationship between mangrove latitude and characteristics of fish communities. (**A**) The relationship between number of fish species and the mangrove latitude. (**B**) The relationship between reef-associated fish and the mangrove latitude. (**C**) The relationship between carnivores and the mangrove latitude. (**D**) The relationship between detritivores and the mangrove latitude. (**E**) The relationship between herbivores and the mangrove latitude. (**F**) The relationship between omnivores and the mangrove latitude. (**G**) The relationship between small-sized fishes and the mangrove latitude. (H) The relationship between medium-sized fishes and the mangrove latitude. (I) The relationship between large-sized fishes and the mangrove latitude. (*: Significant, *p* < 0.05).

**Table 1 biology-11-01696-t001:** Distribution of study areas.

Regions	Areas	Longitude (°E)	Latitude (°N)	Years	Sources
Quanzhou, Fujian	Quanzhou Bay	118.738	24.839	2013–2014	[25]
	Kinmen	118.317	24.433	2011–2012	[31]
Zhangzhou, Fujian	Zhangjiangkou	117.436	23.939	2001–2016	[30,32,33,34,35]
Beihai, Guangxi	Weizhou Island	109.121	21.051	2008	[36]
	Shankou	109.780	21.492	1994–2011	[30,36,37,38,39]
	Yingluo Port	109.784	21.486	1994–1997	[40,41]
Dongxing, Guangxi	Beilun Estuary	107.913	21.573	1990–2008	[26,36]
Fangchenggang, Guangxi	Zhenzhu Port	108.221	21.516	1994–1997	[40]
Qinzhou, Guangxi	Qinzhou Port	108.580	21.889	2011–2012	[42,43,44]
	Maowei Sea	108.580	21.956	2011–2015	[27,28,30,45]
Haikou, Hainan	Dongzhai Port	110.560	20.025	2004–2015	[30,32,46,47,48,49,50,51,52]
Sanya, Hainan	Qingmei Port	109.646	18.233	2019	[53]
	Tielu Port	109.724	18.268	2019	[53]
	Sanya River	109.513	18.266	2019	[53]
Wenchang, Hainan	Wenchang	110.827	19.538	2018	[54]
	Qinglan	110.841	19.560	2015–2016	[55]
Taipei, Taiwan	Tanshui River	121.466	25.161	1989–1990	[56]
	Chu wei	121.400	25.159	1996	[57]
	Shin feng	120.963	24.851	1996–1997	[58]
	Chu Nan	120.860	24.670	1996–1997	[58]
Tainan, Taiwan	Puzih River estuary	120.147	23.485	2008	[59]
	Chiku Lagoon	120.084	23.140	1995–1998	[60]
	Pu tai	120.193	23.475	1996–1997	[58]
	Pei men	120.166	23.314	1996–1997	[58]
	Syh tsao	120.230	23.012	1996–1997	[58]
Hong Kong	Kei Ling Hai Lo Wai	114.289	22.422	2002–2003	[61]
	Nam Wai	114.275	22.354	2002–2003	[61]
	Wong Chuk Wan	114.294	22.392	2002–2003	[61]
	Hong Kong	114.158	22.297	2002–2003	[62]
Eastern Guangdong	Shantou	116.410	23.220	2016–2017	[63]
	Chaozhou	116.929	23.552	2016–2017	[63]
Western Guangdong	Wuli	110.514	21.197	2002–2007	[64,65]
	Hean	110.408	20.696	2002	[64]
	Qishui	109.744	20.776	2002–2007	[64,65]
	Taiping	110.216	21.050	2002	[64]
	Beitan	110.191	20.963	2002–2007	[64,65]
	Tacheng Island	110.444	21.151	2002	[64]
	Fucheng	110.175	20.938	2002–2016	[64,66]
	Jiulongshan	110.341	20.690	2014–2016	[66]
	Leizhou Peninsula	110.185	20.934	2013–2015	[67,68,69]
	Zhenhai Bay	112.424	21.847	1991	[70]
	Wide bay	112.777	21.897	2016–2017	[63]
Zhanjiang, Guangdong	West Zhanjiang	110.132	20.328	2012–2014	[71]
	Xialiu	109.784	21.319	2014–2016	[66]
	Gaoqiao	109.731	21.601	2002–2016	[64,65,66,72,73]
	Xilian	109.893	20.368	2004	[65]
	Liusha	109.892	20.428	2004	[65]
	Zhanjiang	110.900	20.600	2002–2017	[74,75]
	Guanduhekou	110.440	21.163	2009–2010	[76]
Pearl River Delta	Qi’ao Island	113.636	22.419	2016–2017	[63,77]
	Shenzhen Bay	113.965	22.515	1993–2015	[78,79]
	Fukuda	114.030	22.523	1992–2017	[63,80,81]
	Daya Bay	114.641	22.752	2016–2017	[63]
	Lotus Bridge	113.561	22.141	2003–2004	[82]

**Table 2 biology-11-01696-t002:** ANOSIM and SIMPER analysis between groups.

Groups	A	B	C	D
A		1.00	1.00	1.00
B	87.29		0.83	0.92
C	93.38	76.40		0.71
D	89.04	77.85	72.71	

The upper half is the R value of ANOSIM. The lower half is the percentage value of SIMPER.

## Data Availability

Part of the data presented in this study are available in the Appendix A. The remaining data presented in this study are available upon reasonable request from the corresponding author.

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
