# Peer review of "Distribution Pattern of Mangrove Fish Communities in China"

_biology, 2022, doi:10.3390/biology11121696_

Round 1

Reviewer 1 Report

Divide sentence starting however in line 78 and ending at line 82 as it is too long.

Line 133 change for to from

Line 146 change endangered to Conservation status

Line 159 change full length to total length

Line 171 references should be added to support the statement

Lines 232 and 233 Conservation status should be used instead of endangerment

Line 345 add in before low elasticity

Line 401 change additional to addition

Line 443 add the before present study

Line 462 change comma to fullstop after studies

Reviewer 2 Report

REVIEW OF BIOLOGY-2023115

Comments in red are major concerns that needs considerable attention by the authors

In general, this is a reasonably well-written manuscript that deals with the fish community composition and their site-specific variability in multiple mangrove habitats along the coast of mainland China and some of its adjacent island territories. The reporting of fish community characteristics at different study sites is fine and sound. However, the study falls apart when the authors attempt to perform a habitat-wise (site-wise) characterisation these fish communities. This is due to the lack of proper methodological pipeline (or lack of a proper decription of it), entirely ignoring environmental variables in their analyses, overlooking small scale variables and focusing on large scale (latitudinal scale) holistic attributes when attempting to correlate community characteristics to mangrove habitats/sites. Abvoe all, this manuscript does not address these caveats in its Discussion. 

Below, I have provided detaild chapter-wise review and provided rough guidelines to how the authors can improve the manuscript should they decide to do so. 

Despite the present problems, I believe that this work has sufficient merit to be reconsidered after a revision. I appreciate the hard work that the authors put into this and wish them all the best with the revision.

INTRODUCTION

Line 54-55:These habitats provide shelter and feeding and spawning grounds for fish, and they account for half the export of fish with commercial value”: Is this the export value with respect to China? Be specific!

Line 71-73: What about mangrove-associates? (the vegetation associated with mangroves)

Line 79: “…does not make for optimistic prospects”: I am not sure if I understand the message conveyed by this phrase (in the context of the sentence) properly. Consider rephrasing.

METHODS

Line 115: “The tides are varied,…”: This does not read well. Please replace with “The tides vary,…” OR “The tides tend to vary,…

Table 1: Honestly, I do not like the many abbreviations provided to denote different regions. It is too exhausting to read when there are so many non-standard abbreviations and those mean little sense without any visual aid. For a local reader, these sites and regions can be familiar. However, for a reader coming from elsewhere, making sense of 16-odd abbreviated spatial regions (see Table 1) is not easy. I guess the choice to use those may be justified by the authors, as the region names are too long, and it is cumbersome to use those in figures and tables. However, if the authors prefer a broader readership, then, I suggest,

1.       either stick to regional names and drop abbreviations, OR

2.       use abbreviations, but whenever region abbreviation is mentioned in the main text, give the region name within brackets AND add a map showing the demarcations between these regions – OR present those in the existing map AND add a cross reference to that region map when region abbreviations are mentioned in Figures.

There is also some confusion of the terminology. For example, Line 112-113 reads, “To achieve general representativeness of the data and analysis accuracy, we divided the 54 mangrove areas into 16 regions”. Based on this, the abbreviations in Table 1 (1st column) should be named “Regions” – not “Area”. The column, “Sites” (2nd column) should be replaced by “Areas”. Am I correct?

This confusion has bled into the text as well. For example, on the axis label of Figure 2, it says “Study area”, which, based on your definition, should be “Regions”. On the same figure, the caption says, “Relative abundance of fish community composition at the order levels across different 211 mangrove areas”. This once again, has to be “regions” not areas. In Line 200 (Results), it reads, “The number of fish species in each mangrove area was as follows:”. This again has to be “regions”. There are many such confusing instances across the text. When you establish a definition, please stick by it. I, therefore, suggest the authors to read through the entire manuscript and adjust the text so it fits to a universal terminology.

Line 120: Are the numbers referring to rainfall? Please mention the units.

Figure 1: I have two suggestions to improve the presentation. First, the coordinate labels are so small that those are barely readable. Please increase the font size of the coordinate labels. Second, try to reduce overlapping of points by slightly offsetting the positions (lat, lon) of those points.

I also wonder if it will be possible to distinguish these points by geographic regions you established (Table 1). Will the use of colors or labels be practical to denote those regions? Try out and see – if the figure becomes messy and cluttered with labels and colors, then skip this suggestion.

Line 132-141 (2.2.1. List of mangrove fish species): This is where things get a bit cloudy. For a study based entirely on data from published literature + databases, the authors need to describe clearly how the data were extracted. For instance, it is unclear to me whether the authors have searched for specific fish and then classified them according to geographic areas (cf. Table 1) or they focused on geographic areas first and browsed literature to find out what species exist in those. Otherwise, how did the authors know which fish to look for in the database? Therefore, I would like to see a clear methodological description of what systematic steps were followed in the extraction of this data. One other thing that is potentially missing from here is the time range of these records. Assuming that some records go way back in time (say, a few decades), ang given the rapid environmental degradation we see in the past few decades, how confident are authors (1). that those species are still there in the same habitat? – and (2). That those habitats have not changed over time? To account for this, at least include a column to Table 1 indicating the temporal range (e.g., 1990-2015) of the data that you used.

Line 144-145: “and reef-associated fish (RFA) were classified separately for statistical analyses” what does this mean?

Line 144: Please remove the RFA abbreviation. You have more than enough abbreviations in the text already! When you remove it, remove RFA from figures/tables and text and replace with the full term.

Line 148-150: Is this sentence really necessary?

Line 157-161: How were these size ranges established? Did you refer to some previous studies or is it purely arbitrary?

Line 165: Cite R and mention the version number.

RESULTS & DISCUSSION

Line 200-202: This is a sub-optimal way of writing. Since this information is already presented in Figure 1, just write the highlights from that figure. For example, mention which regions had the highest and lowest species richness together with any important information likewise.

Line 206: “…counting for more than 53.2%.”: Please add a cross reference to Figure 2 at the end of this sentence.

Line 220: “Database”: use “database” (small “d”)

Line 209: In this paragraph, add a line or two mentioning whether all orders were found across all habitats. Also, comment on what is the rarest order across these mangrove regions.

Line 219-220: Please re-write this sentence. The way it is written, it gives a strange spin to its meaning!

Line 219-225 & 233-245: Strangely, your region abbreviations have disappeared from this part of the text! This gives the hint to me that these abbreviations can be dropped from the text (may be not from Figures though).

Line 229-230: “The ratio is similar and much smaller than the other two.”: What is meant by the “other two”? Be specific when writing!

Lines 233-245 (3.2.3. Endangerment of fishes in different areas): This section is very important and provides an overview of the potential threats faced by these organisms. However, this is where the time range of these come into play. From what it seems, what is presented here is a sort of “an average” IUCN red-list categorization for the time periods that your records ran through. Have you checked if the red-list categories have changed across the timespan of your records? Let me give you a little bit of context:

Imagine you have a record (database record or literature) for REGION-X and for SPECIES-Y with emerging from 1982 (40 years ago), saying “we found species-Y on region-X in our 1982 field survey”. When you run an IUCN red-list category search for species-Y today, how prudent is it to compare with a datapoint obtained 40 years earlier?

Also, there can be seasonal influences (time-of-year effect) or tidal influences (time-of-day effect) on the species diversity and richness because of these data coming from various years, possibly sampled in various seasons and sampled in various times of day. These effects should be addressed in the discussion.

Figure 6: Here, there seems to be a classification based on mangrove extent (size). I didn’t find any information about a classification of study locations based on the mangrove extent in the Methods section. If this is the case, then that mangrove classification based on its extent must be mentioned in the Methods!

Also on Figure 6, the r and p values are overly precise. Use maximally 2-3 decimal places. If p-values are too small, use notations such as, p < 0.05 or p < 0.01 etc. Exact p-values are not always needed, particularly when they are very small.

Also, keep a space between test statistic, equals sign and the value. E.g., “p = 0.039” NOT “p=0.039”!

Also, what does the unit “size/lg hm2” refer to? Is it square hectometer? Why not stick with standard (SI) units? Refer to journal guidelines and stick to standard form of area units. What is “size/lg”?

In Figure 6 caption, the phrase “proportion of […..] in different mangroves” appear many times. What is meant by “different mangroves”? Shouldn’t it really be termed, “the relationship between […..] and the mangrove extent” or something likewise?

Panels G, H & I: Rather than testing for the three size groups separately, the authors should have tested the size of fish (no pre-defined size groups) against the mangrove extent. That may have given a different pattern. Do this analysis separately and see if you get any different output.

NOTE: These adjustments are more or less apparent to Figure 7 too. Compare, see what is applicable and fine-tune the Figure 7 accordingly.

Line 287: Due to the re-classification of study sites based on the mangrove extent (as shown in Fig. 6 x-axis and mentioned above) the term, “mangrove area” in the context of this correlation test is ambiguous. This is because, in the Methods, it was written, “To achieve general representativeness of the data and analysis accuracy, we divided the 54 mangrove areas into 16 regions”. So, for much of the text, the authors (erroneously) referred to “mangrove areas” to denote the spatial regions (abbreviations in Table 1). But now, with mangrove sites reclassified based on their extent, what does this “mangrove area” on Line 287 and onwards refer to? I am sorry, but this seems like a horrendous confusion of terms!

Line 289: “found showed”: pick one term and remove the other!

Section 3.4 (correlation analysis): Comparing mangrove extent to the fish community characteristics is nice. However, is it sensible to make the same analysis along a mere 6.5 degrees latitudinal gradient? Although this has been mentioned somewhat in the Discussion, in my view, there are many other variables that these fish community characteristics should have been compared against. For example, the authors should have extracted the temperatures (hinting seasonal variations), tidal levels & salinities (hinting diel variations), depths, mangrove community compositions (at least the dominant mangrove taxa), extent of human influences at each mangrove site/region, pollutant discharge, urbanization (e.g., distance to the nearby city), canopy cover etc. This is because one cannot simply explain the fish community dynamics based on mangrove extent and its limited latitudinal variability. The mangrove extent and latitude are two greater variables at play, while the existence, diversity and composition of fish within a mangrove habitat can mostly be governed by variables operating on a local scale (e.g., temperature, bottom depth, pollutants, habitat quality [e.g., root system network], tidal regimes, ocean current patterns, time of the day, day of the season etc..). Therefore, in my view, these key small-scale variables are entirely ignored and their potential influence on the fish community dynamics are overlooked in this study.

To get meaningful insights about fish community comparisons/trends, it could be very helpful if the authors re-focus their attention at least a few of the selected smaller-scale variables mentioned above (and there are many that are not mentioned also). The authors could approach this by extracting additional data (e.g., site-and time-specific environmental data from reanalysis archives or past studies) & re-performing the correlation analysis with more meaningful variables. However, this will be a humongous task as the fish records could be scattered over a broad range of times and looking for environmental data for those times could be very exhausting or near-impossible. Therefore, I do not push the authors to follow this approach – but I would like if the co-authors can openly discuss and agree on if such additional data can be extracted and more analyses can be performed. This decision, therefore, is up to the authors to make.

However, if the authors decide not to re-perform the statistical analyses with more meaningful (and more influential) variables, then, the Discussion must be sufficiently self-critical of the potential weaknesses of the methods followed (see above comments where I refer to problems with the temporal scattering of the data). Also, the second correlation analysis with latitudes (Figure 7) should be dropped from the main text (and kept in an appendix). At its current state, the Discussion chapter does not look sufficiently inwards – nor have the self-critique that is much needed. I would, therefore, prefer if the authors focus more Discussion space on describing why certain patterns emerge from the data (clustering patterns or correlations) and guide the readers with statements of caution, while pointing to potential weaknesses of the methodological approach of this study.

<end>

Reviewer 3 Report

The authors present an interesting manuscript on mangrove fish diversity in China. The bright side of the manuscript is that it provides some useful practical details on the related topic. In this context, the study contributes to the understanding of mangrove fish diversity in China. The authors explained their aims, methods, and results but some sentences of the paper manuscript are not easy to understand. Moreover, there are some missing points in the manuscripts. For example, statistical methods and results should be clearly described. Additional references are needed in several parts of the manuscript (mentioned below). Therefore, I would like to make some suggestions to improve the quality of the paper as below:

Abstract

Line 31: Please specify the “data”. Which data sources were used?

Introduction

Lines 71-74: “It is worth noting that restoration efforts of mangroves do not only concern mangrove trees but should also be directed to integrate the important ecological functions of fish, water birds, and benthic animals [15-17].” Please rephrase this sentence.

Lines 78-82: “However, the current status of mangrove fish communities does not make for optimistic prospects, owing to overexploitation and disproportionate targeting of fish at high trophic levels by the growing human population, which will affect fish population structures and life history process” The sentence is too long. Please rephrase this sentence.

Materials and Methods

Lines 133-134: “Statistics of fish species for historical research in various mangrove areas in China were used.” More information is needed here. Please clarify the Statistics of fish species. Morphometrical characters that were measured by previous researchers were used for statistical analyses or statistics of previous studies were used or something else?

Line 143: fish -> fish species

Line 148: fish -> fish species

Line 151: 2.2.4. Endangered - > Conservation status of fish species

Line 152: endangerment status - > Conservation status

Lines: 158-161: More information is needed for the body size of fish species. Did authors use only male individuals for body size? Did authors use unsexed individuals? Please clarify.

Line 161: A reference is needed here.

Line 162-183: Authors should explain the statistical analyses in more detail. PRIMER, R and SPSS were used but which software was used for which analyses?  

Line 171: “It is generally believed that stress < 0.2 has explanatory significance”. A reference is needed here.

Line 179: A few words about fish community characteristics would be better here.

Line 183: A reference is needed here.

Results

Line 190: The next most abundant -> The second most numerous

Line 205: all -> each

Line 232: 3.2.3. Endangerment of fishes in different area -> Conservation status of fishes in different areas

Line: 233: After identifying the endangerment status of fish in the Fishbase Database -> After identifying the conservation status of fish species in the Fishbase Database

Lines 234-235: endangered on some level -> threatened

Line: 238: occurred -> detected

Lines 247-250: Small-sized fishes, medium-sized fishes and large-sized fishes.

Line 251: Larger legends would be better for Figure3.

Line 286: “The total number of fish species was strongly positively correlated with mangrove area (r = 0.76, p < 0.01)”. I think, “positively correlated with mangrove area” is enough since r is 0.76, not so strong.

Line 287: “mangrove area” size of the mangrove area or habitat of the mangrove area? Please clarify.

Line 289: found -> results

Line 307: not strongly correlated with mangrove latitude -> not correlated with mangrove latitude

Lines 309 and 317: A larger figure would be better for Figures 6 and 7.

Discussion

Line 356: Did any specific pollutants that harm fish species detect in previous studies? If yes, a few words about that would be better here.

Lines 371-373: “It is worth noting that according to the investigation on habitat types of mangrove fish in China, reef fish accounted for 29% of the total fish diversity, and for 45% in Hong Kong mangrove areas.” Please rephrase this sentence.

Line 374: small fish -> small-sized fishes

Line 444: weakly and non-significantly correlated -> not correlated

Conclusion

Lines 471-480: the limitations of the study should be described.

Round 2

Reviewer 2 Report

I thank the authors for investing significant time in revising the manuscript, which now looks more readable and scientifically sound. I agree with their justifications where additional data/analyses cannot be added/performed at the current state. 

I, therefore, do not have any major concerns about this manuscript. 

Nonetheless, four minor adjustments can be done. This will barely take 30 minutes.

1. Add the unit (East orNorth) to the column names of Table 1 as, "Longitude (°E)" and "Latitude (°N)"

2. In the Longitude and Latitude columns, please stick to a universal (2, 3 or 4) decimal places in all rows. Currently, the decimal precision differ between rows.

3. Since the tabulated Latitude and Longitude values are in Decimal Degrees (e.g., 17.263 °N), if the authors are able, please change the units of the coordinate grid to Decimal Degrees as well. Curently, it is at DD.M.S.

4. Finally, since the authors have decided that it is impracticable to add more fine spatial scale data/analyses to the manuscript (which is perfectly fine), they can bring back the previous Figure 7 to the main text (which is now in Supplement, as per my comment in previous round). Please change the cross reference to this figure between Lines 337-346 (Results) and elsewhere in the Discussion, if they bring it back to the main text. I apologise if this caused the authors to do annoying "back-and-forth" type of work.

I wish the authors all the best in their future scientifc work! 

Reviewer 3 Report

The authors present an interesting manuscript on mangrove fish diversity in China. The bright side of the manuscript is that it provides some useful practical details on the related topic. In this context, the study contributes to the understanding of mangrove fish diversity in China. The authors improved the manuscript with the previous comments. Missing points in the manuscripts mentioned before were explained. However, I have a few suggestions to improve the quality of the manuscript.

Line 31: database -> two databases

Line 46: fish -> fishes

Line 151: The conservation status of fish -> The conservation status of each fish species

Line 393: The 16 mangroves regions -> The 16 mangrove regions

Lines 504-508: “Only latitude and area of two variables was analyzed, and ignore the small-scale variables for the potential effect of fish communities. Examples include temperature, tides levels, salinities, depths, mangrove community composition, human influences, pollutant discharge, and urbanization etc.”. These sentences are not easy to understand so they should be rewritten. Such a sentence (or a similar sentence) would better fit here “Only two variables, which are latitude and mangrove, were analyzed and small-scale variables for the potential effect of fish communities including temperature, tides levels, salinities, depths, mangrove community composition, human influences, pollutant discharge, and urbanization were ignored”.
